# Major Allele Frequencies in *CYP2C9* and *CYP2C19* in Asian and European Populations: A Case Study to Disaggregate Data Among Large Racial Categories

**DOI:** 10.3390/jpm15070274

**Published:** 2025-06-27

**Authors:** Horng-Ee Vincent Nieh, Youssef Malak Roman

**Affiliations:** 1Duke Raleigh Hospital, A Campus of Duke University Hospital, 3400 Wake Forest Rd., Raleigh, NC 27609, USA; 2L.S. Skaggs College of Pharmacy, Idaho State University, 1311 E Central Dr, Meridian, ID 83642, USA; 3Boise Veterans Affairs Medical Center, 500 W Fort St, Boise, ID 83702, USA

**Keywords:** Asian, European, population subgroups, racial categorizations, CYP2C9, CYP2C19, pharmacogenomics, precision medicine, diversity

## Abstract

CYP2C9 and CYP2C19 are major CYP450 enzymes that heavily influence the hepatic metabolism and bioactivation of many medications, including over-the-counter and narrow therapeutic index drugs. Compared to the wild-type alleles, genetic variants in either gene could potentially alter the pharmacokinetics of widely used medications, affect the desired therapeutic outcomes of a drug therapy, or increase the risk of undesired adverse events. The frequency of genetic polymorphisms associated with CYP450 enzymes can widely differ across and between racial and ethnic groups. This narrative review highlights the differences in *CYP2C9* and *CYP2C19* allele frequencies among European and Asian population subgroups, using published literature. Identifying the substantial differences across European and Asian populations, as well as within Asian subgroups, indicates the need to further scrutinize general population data. Clinical scientists and healthcare providers should advocate for more inclusive clinical pharmacogenomic data and racially and ethnically diverse pharmacogenomic databases. Clinical trials of limited racial and geographical diversity may not necessarily have strong external generalizability for all populations. Furthermore, clinical trials that designate an all-inclusive Asian population consisting of multiple ethnicities may not be adequate due to the perceived genetic differences among Asian subgroups. Gravitating towards a more comprehensive approach to utilizing pharmacogenomic data necessitates granular population-level genetic information which can be leveraged to improve how drug therapies are prescribed, achieve health equity, and advance the future of precision medicine.

## 1. Background

The Cytochrome P450 enzymes are responsible for the hepatic biotransformation or activation of medications and endogenous compounds in the human body. As members of the Cytochrome P450, CYP2C9 and CYP2C19 (CYP450 family 2 subfamily C member 9 and 19) enzymes have significant roles in several drug therapies. Together, they metabolize approximately 20–30% of all drugs. It is estimated that CYP2C9 metabolizes 15% and CYP2C19 metabolizes 8–10% of all medications in current use [1].

CYP2C9 is a phase I drug-metabolizing enzyme responsible for the biotransformation of widely prescribed drugs [2]. Genetic variability in the *CYP2C9* gene can alter the amino acid sequence of the encoded protein, which can change substrate recognition sites and therefore the overall enzymatic activity [3]. CYP2C19 is another important drug-metabolizing enzyme with well-characterized phenotypes—normal, intermediate, poor, and ultrarapid metabolizers [4]. Generally, patients carrying genetic variations leading to reduced or no enzymatic activity are considered poor metabolizers and may exhibit unwanted adverse drug reactions compared with normal metabolizers. On the other hand, individuals who carry genetic variations that are associated with increased metabolic activity may compromise the desired therapeutic response due to increased enzymatic activity [5].

Most CYP2C9 and CYP2C19 drug substrates are weak acids and include both over-the-counter and prescription medications [3]. Both Table 1 and Table 2 provide a summary list of common prescription drug labels that are metabolized by CYP2C9 and CYP2C19. The PGx levels of recommendations reflect whether a requirement or recommendation for genetic testing exists, and whether an actionable dose adjustment, contraindications, or therapeutic interchange in the approved drug label is indicated by the respective regulatory agency, including the U.S. Food and Drug Administration (FDA), European Medicines Agency (EMA), Swissmedic (SM), or Health Canada (HC). The clinical annotations of drugs by major pharmacogenomics clinical guidelines provide additional guidance on treatment selection, dose modifications, and monitoring parameters whenever patient genetic information is available. Drug label and clinical annotation information were extracted from the Pharmacogenomics Knowledgebase (PharmGKB) website [6,7].

The pharmacogenomic label information provided by these international drug agencies (FDA, EMA, SM, and HC) can assist the clinical decision-making process and represents a step in the right direction toward realizing the benefits of precision medicine [6,7]. However, the available prescribing information may not always be adequate, partly due to the lack of diversity in the patient populations to inform a genetically based guideline or the generalizability of drug efficacy [8,9]. A well-designed clinical trial will include multiple sites with diverse populations worldwide, but even then, it can only represent a few subgroups of the global population to which the drug will be distributed [10,11]. Outcome data can also be compromised by information health disparities, especially when ethnically diverse subgroups are aggregated into a single population [12]. A common example of this practice is when Asian and Pacific Islander populations are underrepresented or consolidated into one all-inclusive population group. In 2024, the United Nations estimated that 4.3 billion people live in Asia and the Pacific Islands, accounting for 60% of the global population. The region is home to the world’s two most populous countries – China and India [13]. Despite their large population size, the representations of these populations in global drug development remain limited.

Historically, individuals of European ancestry have been more prominently represented in clinical trials than those of Asian descent, despite a recent 67% increase in Asian representation over the past two decades [14,15]. Although race-specific dosing recommendations have been established for some Asian individuals, such as lower dosing regimens for rosuvastatin and warfarin, the significant underrepresentation of certain population subgroups in clinical trial data can lead to major public health concerns and exacerbate health disparities [16,17,18,19]. Inadequate representation may result in investigational new drugs producing subtherapeutic plasma concentrations or posing a higher risk of adverse drug reactions in the very populations excluded from or underrepresented in the clinical development process. Previous studies have even identified variability among subgroups within the broader Asian population, further underscoring the importance of increasing racial and ethnic diversity in clinical pharmacogenomic research [16,17].

Genetic variability in the *CYP2C9* and *CYP2C19* genes among Asian populations is a critical area of discussion due to its significant implications for drug metabolism and personalized medicine. Select variants in *CYP2C9* and *CYP2C19* are more prevalent in East Asian populations compared to European populations, leading to reduced enzyme activity and altered drug pharmacokinetics [20]. For *CYP2C9*, the *CYP2C9**2 and *CYP2C9**3 alleles have been identified as the most common decreased-function alleles in individuals of European ancestry [21]. In contrast, the *CYP2C9**5, *6, *8, and *11 alleles are most frequent in those of African ancestry [22]. The most common non-functional *CYP2C19* allele, *CYP2C19*2*, has been identified to have a greater frequency among Asian populations than European populations [23]. This difference in allele frequencies exemplifies the concept of genetic polymorphism and its interpatient variability among different populations. Derived from the germline cells, polymorphisms are essential in creating genetic diversity but can also affect an individual’s innate ability to metabolize or activate certain drugs via the CYP450 enzymes. Altered enzyme activity will subsequently affect the hepatic clearance, bioavailability, and elimination half-life, and most importantly, how an individual will respond to a particular medication. A summary of the potential *CYP2C9* and *CYP2C19* diplotypes and their corresponding phenotypes is shown in Table 3 and Table 4. As such, incorporating pharmacogenetic testing into clinical decision making can optimize dosing, minimize adverse drug reactions, and improve therapeutic outcomes in these populations [24,25,26]. This also underscores the importance of ethnicity-specific guidelines and further research into the clinical translation of CYP450 genetic variability in diverse ancestral populations.

## 2. Objective

The objective of this narrative review is to evaluate the frequencies of major *CYP2C9* and *CYP2C19* alleles among some of the largest Asian population subgroups in the United States compared to those of European subgroups. Acknowledging the differences between the European and Asian populations and further within the Asian subgroups can improve how future clinical trials recruit participants, scrutinize population-level racial categorization, enhance the collection of diversity biomarkers in clinical trials, increase the awareness of healthcare providers about medications impacted by highly polymorphic drug-metabolizing encoding genes, and eventually improve therapeutic outcomes for underrepresented populations.

## 3. Methods

A PubMed search was conducted to identify articles containing pharmacogenomic data on European and Asian populations. Search terms included “Cytochrome P450,” “allele frequency,” “CYP2C9,” “CYP2C19,” “variability,” “polymorphism,” and “pharmacogenomics.” Ethnicities included “Asian,” “Chinese,” “Filipino,” “Hmong,” “Indian,” “Japanese,” “Korean,” and “Vietnamese.” Articles were included in this review if they contained at least one of the population subgroups of interest and the allele frequency was stated clearly or could be calculated from the study population using available genotype counts and sample sizes. Articles that did not explicitly list an allele frequency in the results table but instead mentioned the percentage in the Results or Discussion sections were excluded. Included articles also addressed at least one of the CYP2C9*2, **3*, **5*, **8*, and **11* alleles, or *CYP2C19*2*, **3*, and **17* alleles, and all relevant data were included when available. Only articles published in English were reviewed. The Clinical Pharmacogenetics Implementation Consortium (CPIC) Guidelines, PharmGKB, and Dutch Pharmacogenetics Working Group (DPWG) resources were used to cross-reference pharmacogenomic information and drug dosing recommendations.

## 4. Results

This narrative review included 34 articles regarding data from European, Chinese, Filipino, Hmong, Indian, Japanese, Korean, and Vietnamese populations to compile the frequencies of the major alleles in both *CYP2C9* (Appendix A) and *CYP2C19* genes (Appendix A). Notably, the literature search yielded more published data from European, Chinese, Korean, and Japanese populations, while there were fewer articles addressing Filipino, Hmong, and Vietnamese populations. CYP2C9-related data were more robust for the *CYP2C9*2* and *CYP2C9*3* alleles than the *CYP2C9*5*, *8, and *11 alleles for all subgroups, consistent with disproportionate allele frequencies in Blacks or African Americans. Additionally, more publications addressed the non-functional *CYPC19*2* and *CYP2C19*3* alleles than the increased-functional *CYP2C19*17* allele. At least two studies with data on at least two alleles were included for each population subgroup. The frequency ranges of the *CYP2C9* and *CYP2C19* alleles that were reviewed are summarized in Figure 1 and Figure 2.

### 4.1. CYP2C9

After analyzing the aggregated data, the European allele frequency ranges were 9.9% to 15.7% for *CYP2C9*2* and 5.3% to 9.8% for *CYP2C9*3*. In comparison, the Asian allele frequency ranges were 0% to 4% for *CYP2C9*2*, 0.5% to 18.9% for *CYP2C9*3*, 0% for *CYP2C9*5*, 1.8% for *CYP2C9*8*, and 0% to 0.05% for *CYP2C9*11*. The *CYP2C9* allele frequency ranges of the European population and each Asian population subgroup are shown in Figure 1. The complete data of the populations and subgroups are compiled in Table 5.

European subgroup data identified the *CYP2C9*2* frequency to be 14.7% for Croatian, 12.1% for Danish, 9.9% for Norwegian, 11.3% for Romanian, 11.7% for Serbian, 15.6% for Spanish, and 15.7% for the combination of Spanish and Northern Italian populations. The CYP2C9*3 frequencies were 7.6% for Croatian, 5.3% for Danish, 6.5% for Norwegian, 9.3% for Romanian, 8.1% for Serbian, 9.8% for Spanish, and 7.8% for the combination of Spanish and Northern Italian populations (Table 5).

Data derived from the Asian populations showed a lower frequency range for CYP2C9*2 than the European populations. No Asian subgroup had a CYP2C9*2 frequency greater than 0.5%, except the Indian subgroup at 4%. The range for the CYP2C9*3 allele frequency was wider in comparison, with the Hmong population representing greater frequencies of 16.6% to 18.9%, which heavily influenced the overall Asian range. Additionally, none of the Asian subgroups had a CYP2C9*3 frequency greater than 9.0%, other than the Hmong subgroup at 16.6 to 18.9%. Within the same population subgroup, the *CYP2C9*3* frequency showed the greatest variability in Chinese, ranging from 2.0 to 9.0%.

A comparison of the European and Asian data shows a significant difference in the allele frequency for *CYP2C9*2*. A European individual is more likely to express the *CYP2C9*2* allele than an individual of Asian descent, based on the studies evaluated in this review. The lower limit of the range in the European data (9.9%) remains significantly greater than the upper limit of the range in the Asian data (4.0%). The data also show that a European individual is more likely to express the *CYP2C9*3* allele unless the Asian individual is of Hmong or Indian descent. The *CYP2C9*3* frequency of the Hmong subgroup is significantly greater than the other European and Asian populations.

### 4.2. CYP2C19

The range of the CYP2C19 allele frequencies of the European and Asian populations subgroups is shown in Figure 2 and summarized in Table 6. The European data showed a frequency range of 11.1–16.3% for *CYP2C19*2*, 0% for *CYP2C19*3*, and 19.6–25.5% for *CYP2C19*17*. Subgroup data for the CYP2C19*2 frequencies were 15.0% for Danish, 15.2% for German, 13.1% for Greek, 11.1% for Italian, 15.2% for Norwegian, and 16.3% for Serbian populations. There was no occurrence of the *CYP2C19*3* allele in European subgroups. The *CYP2C19*17* subgroup frequencies were 23.9% for Croatian, 20.1% for Danish, 25.5% for German, 19.6% for Greek, 22.0% for Norwegian, and 22.2% for Serbian populations.

The overall Asian population data showed a wider range for each of the *CYP2C19* variant allele frequencies than Europeans. The Asian data showed a frequency range of 20.5–53.8% for *CYP2C19*2*, 0–15.6% for *CYP2C19*3*, and 0.5–17.9% for *CYP2C19*17.* There was a greater frequency for the non-functional alleles *CYP2C19*2* and *CYP2C19*3* in all Asian subgroups than in the European population. Among the Asian subgroups, the Chinese (24.9–45.5%) and Indian (22.0–41.7%) subgroups demonstrated the widest range of *CYP2C19*2* expression. The Hmong (0–0.3%) and Indian (0–1.2%) subgroups demonstrated the lowest expression of *CYP2C19*3*. The *CYP2C19*17* increased-function allele was significantly more common in the European population at 19.6–25.5%.

The data show marked differences in the allele frequency for the *CYP2C19*2*, **3*, and **17* alleles. An Asian individual is much more likely to have either a non-functional *CYP2C19*2* or *CYP2C19*3* allele than someone of European descent. The maximum of the allele frequency range in the European data is lower than the lowest value in any of the Asian subsets for *CYP2C19*2.* There were no data to suggest that any subgroup expressed the *CYP2C19*3* allele in the seven studies that were reviewed for populations in Europe. Aside from the Indian subgroup (10.2–17.9%), no other Asian subgroup expressed the *CYP2C19*17* allele greater than 2.1%, which is over ten times less than that of the European average of 22.3%.

## 5. Discussion

In this review, the Asian subgroups included Chinese, Filipino, Hmong, Indian, Japanese, Korean, and Vietnamese populations. Figure 1 shows that the *CYP2C9***2* and *CYP2C9*3* subgroup frequencies are relatively similar based on their geographic proximity. For example, the Vietnamese and Filipino subgroups have similar ranges but differ from that of the Indian subgroup. The Chinese, Japanese, and Korean subgroups also appear to have similar frequencies for the *CYP2C9* variants.

Similar trends in *CYP2C19* allele frequencies were also observed (Figure 2) in both Asian and European groups. Geographical differences among European countries remained marginal, with relatively comparable *CYP2C19* allele frequencies. Unlike Asian subgroups, the European population subgroups demonstrated less variability in the *CYP2C19* allele frequencies. Based on the studies reviewed, the *CYP2C19*3* allele was not identified in any of the European subgroups but was moderately prevalent in each of the Asian subgroups—the Japanese subgroup had the highest frequency, ranging from 9.1% to 15.6%. The Hmong and Filipino subgroups exhibited greater frequency for the *CYP2C19*2* allele than all other Asian subgroups. Additionally, *CYP2C19*3* allele frequencies were relatively rare in the Hmong and Indian populations. These trends may also highlight the genetic differences that may result from ancestral migration. The decreased allele frequency variabilities in European subgroups may suggest that aggregating Asian population data may be more problematic than aggregating European population data [21]. This difficulty could be due to increased population diversity and the underrepresentation of ethnographic groups with distinct genetic profiles. Additionally, it was suggested that the low frequency of the *CYP2C19*17* in a multi-ethnic population subgroup in Southern China could be attributed to evolutionary adaptations to the low-oxygen, high-altitude environment [61]. However, a link has not been fully established.

The Chinese subgroup demonstrated the greatest variability in the *CYP2C9*3* and *CYP2C19*2* alleles compared to any other population subgroups included in this review. This can be attributed to the ethnic diversity within the Chinese population, which includes the Han Chinese majority, 55 officially recognized minority groups, and additional unrecognized ethnic groups [62]. When reviewing the *CYP2C19* data, two articles reviewed specific data for individual Chinese subgroups and identified some variabilities that exist within the country [44,46].

The Hmong subgroup demonstrated a *CYP2C9*3* allele frequency that was vastly different from all other Asian subgroups in this analysis. This could be shaped by the unique history of the tight-knit Hmong communities that originally settled in present-day Vietnam, Cambodia, and Laos and then relocated to refugee communities across the United States [63]. The isolated and displaced history of the Hmong population may have led to genetic compositions that significantly differed from their geographical neighbors. Due to the scarcity of data, one of the studies for Hmong data derived *CYP2C9* data from Hmong populations living in the United States [50]. Another study used for *CYP2C19* data utilized data from the Hmong population in Vietnam [40].

The Indian subgroup exhibited distinct frequency ranges for *CYP2C9*2*, *CYP2C9*3*, and *CYP2C19*17* compared to other Asian subgroups. In addition to its isolated geographic location from other Asian subgroups, India has a complex history of migration, with ancestral contributions from Eurasia, Southeast Asia, Africa, and Europe, thereby enriching its genetic diversity [64]. India’s rapid population growth to become the most populous country in the world in recent years could also contribute to its distinct genetic profile. Notably, the Indian subgroup’s CYP2C9*3 frequency range (8.0–9.0%) is more comparable to European subgroups (7.6–9.8%) than other Asian subgroups. Similarly, the *CYP2C19*17* range (10.2–17.9%) aligns more closely with the European population (19.6–25.5%).

Based on the *CYP2C19* data, we observed more non-functional *CYP2C19*2* and *CYP2C19*3* alleles in the Asian subgroups compared to the Europeans. Additionally, there was a greater frequency of the increased-function *CYP2C19*17* allele in each of the European subgroups than in any of the Asian subgroups. The frequency percentages of the general European and Asian populations in our review were very similar to those stated in the CPIC Guidelines for CYP2C19 and proton pump inhibitor (PPI) dosing, which estimated a *CYP2C19*2* allele frequency of 15% in Europeans and 25–30% in Asians and a *CYP2C19*17* allele frequency of 20% in Europeans [23]. These trends may suggest that medications that are metabolized by CYP2C19, such as PPI, serotonin reuptake inhibitors, and tricyclic antidepressants, are more likely to have increased bioavailability in Asian individuals and lower bioavailability in European individuals. Conversely, medications that are bioactivated by the CYP2C19 enzyme, such as clopidogrel, may be less likely to reach therapeutic levels in individuals of Asian descent.

Inter-ethnic variations for CYP2C9 allele frequencies may impact how different populations metabolize common and widely used drugs such as nonsteroidal anti-inflammatory drugs (NSAIDs), including ibuprofen and celecoxib [65]. Similarly, *CYP2C9* genetic variability can have a profound impact on narrow therapeutic index drugs, like warfarin [22,66]. For NSAIDs, both the *CYP2C9*2* and *CYP2C9*3* alleles exhibit a decreased metabolic rate compared to the *CYP2C9*1*. The Clinical Pharmacogenetics Implementation Consortium (CPIC) defines the *CYP2C9*1/*2*, *CYP2C9*2/*2*, and *CYP2C9*1/*3* diplotypes as intermediate metabolizers and the *CYP2C9*2/*3* and *CYP2C9*3/*3* diplotypes as poor metabolizers, warranting dose modifications for NSAIDs metabolized by CYP2C9 [66]. CYP2C9 is a major contributor to warfarin metabolism and could partly explain the interindividual variability of the initial dosing requirement strategy. CYP2C9 is the primary metabolizer of the more potent S-warfarin enantiomer. Previous studies have estimated that *CYP2C9*2* decreases warfarin metabolism by 30–40%, while *CYP2C9*3* decreases warfarin metabolism by 80–90%. Consequently, inheriting one or two copies of either allele can increase warfarin bioavailability and bleeding risk during warfarin therapy and result in lower dosing strategies to achieve the desired therapeutic benefit [22]. Indeed, information about CYP2C9, VKORC1, and CYP4F2 phenotypes could optimize the initial dosing of warfarin and achieve the target International Normalized Ratio (INR).

Variations in CYP2C19 allele frequencies can also influence the metabolism of both over-the-counter and prescription drugs. PPIs are common drugs that irreversibly inhibit gastric acid production via parietal cells. The CYP2C19 enzyme is estimated to be responsible for nearly 80% of the metabolism of first-generation PPIs, such as omeprazole and pantoprazole. Second-generation PPIs are less dependent on CYP2C19 metabolism. CPIC Guidelines for CYP2C19 and PPI dosing identify the *CYP2C19*1/*1* diplotype as a normal metabolizer. Possessing the *CYP2C19*17* increased-function allele can identify the individual as a rapid metabolizer (*CYP2C19*1/*17*) or ultrarapid metabolizer (*CYP2C19*17/*17*). A non-functional allele, such as *CYP2C19*2* or *CYP2C19*3*, can lead to an intermediate metabolizer (*CYP2C19*1/*2* or *CYP2C19*1/*3*) or poor metabolizer (*CYP2C19*2/*2*, *CYP2C19*3/*3*, or *CYP2C19*2/*3*). For omeprazole, lansoprazole, pantoprazole, and dexlansoprazole, the CPIC guidelines recommend a 100% increase in the initial dosing for ultrarapid metabolizers. Additionally, intermediate and poor CYP2C19 metabolizers can consider a 50% daily dose reduction for chronic therapy [23]. Conversely, clopidogrel is a prodrug that requires hepatic biotransformation to its active form. The CYP2C19 enzyme is responsible for 44.9% of the first activation step and 20.6% of the second activation step [5]. Consequently, intermediate and poor CYP2C19 metabolizers are at a greater risk for therapeutic failure due to reduced active metabolite formation. For acute coronary syndrome and/or PCI indications, the CPIC guidelines strongly recommend intermediate metabolizers to avoid the standard dose of clopidogrel if possible and to use prasugrel or ticagrelor as alternative therapies. CYP2C19 poor metabolizers should avoid clopidogrel entirely if possible [24,58].

While the response to treatment is multifactorial, it is estimated that 20–30% of the variability in the therapeutic effect is due to genetic polymorphisms [21]. As such, it has become common for medication-prescribing information to include pharmacogenomic data, particularly dose recommendations based on drug metabolism status. To this end, pharmacogenomic testing can assist therapeutic dose monitoring (TDM) by explaining therapeutic failure, optimizing drug selection, and improving patient-centered therapy regimens [67]. When used synergistically, pharmacogenomics testing and TDM can offer enhanced insights into patient care, especially when pharmacogenomics results are inconclusive and/or when the patient is experiencing multiple comorbidities and recurrent therapeutic failures.

The *CYP2C9*2* is associated with 50–70% reduction in enzymatic activity, and *CYP2C9*3* can result in 75–99% reduction in metabolism of CYP2C9 substrates (Table 3). In the case of phenytoin, a significant reduction in the metabolizing effect of CYP2C9 is associated with an increased risk of neurotoxicity. Data have indicated that up to 40% of patients in Southern Europe and the Middle East may benefit from a 25–50% reduction in maintenance doses [21]. These findings underscore the importance of moving beyond broad racial or ethnic categories, which can mask critical genetic diversity within populations. Precision medicine requires a granular understanding of allele frequencies and drug responses in specific subpopulations [68]. A broader study that explored the efficacy of preemptive pharmacogenomic testing on all newly prescribed drugs with actionable DPWG recommendations found that clinically relevant adverse drug events could be reduced by up to 30% [26].

Although numerous case studies support the relevance of individual genetic variation in informing treatment strategies, this review focuses specifically on interethnic differences in allele frequencies of *CYP2C9* and *CYP2C19*—two key drug-metabolizing enzymes with significant implications for managing prevalent and chronic conditions. Current research underscores substantial variability in the distribution of these alleles across Asian subpopulations, highlighting the critical need for diversity in both clinical and genomic studies [8]. Importantly, aggregated racial categories such as “Asian” often obscure meaningful genetic differences among subgroups, including East Asian, South Asian, and Southeast Asian populations [69]. This oversimplification may perpetuate inaccurate assumptions and hinder the advancement of truly personalized medicine. Relying on broad racial classifications risks reinforcing the problematic use of race as a biological proxy rather than focusing on specific, actionable genetic markers [70]. Clinical pharmacogenetic guidelines frequently provide general recommendations based on pan-ethnic categories, potentially neglecting subgroup-specific risks and responses to therapy. In a highly globalized and mobile world, pharmacogenomic databases should also strive for a more inclusive and diverse representation of countries and population subgroups. We advocate for disaggregated data analysis and recommend that healthcare providers incorporate preemptive pharmacogenetic testing, especially when prescribing medications metabolized by polymorphic enzymes like *CYP2C9* and *CYP2C19*, to optimize drug efficacy and safety on an individual level.

## 6. Limitations

The literature search revealed a limited number of studies published within the past five years focusing specifically on the Hmong and Vietnamese subgroups, indicating a persistent gap in pharmacogenomic research for these populations. Similarly, recent data on European subgroups were scarcer than anticipated, particularly outside of Southern Europe. There were no data regarding European allele frequency for *CYP2C9 *5*, **8*, and **11* in the reviewed articles. To enhance the comprehensiveness of the narrative review, broader search queries were employed, which enabled the inclusion of additional studies relevant to the cytochrome P450 enzyme family and Asian populations more generally. Despite this, only one study was identified that specifically addressed the *CYP2C9 *5*, **8*, and **11* alleles, and it included data exclusively from Chinese, Japanese, and Korean subgroups [33]. Notably, much of the available data on European populations originated from Southern European countries bordering the Mediterranean, many of which have historically been underrepresented in large-scale clinical trials. This trend reflects a broader issue of uneven representation in pharmacogenomic research, further emphasizing the need for expanded, subgroup-specific investigations to support equitable and evidence-based clinical decision-making.

## 7. Conclusions

This narrative review highlights the significant and quantifiable differences in *CYP2C9* and *CYP2C19* allele frequencies among European and Asian populations, while underscoring the importance of further disaggregating data to reveal meaningful genetic distinctions among Asian subgroups. The observed variability in allele frequencies likely reflects differences in optimal drug dosing strategies, which can have critical implications for efficacy and safety across diverse populations. Beyond these enzymes, other cytochrome P450 variants and pharmacologically relevant genes may also exhibit population-specific patterns that warrant further investigation. To enhance the applicability of pharmacogenomic research, healthcare providers and researchers must advocate for more inclusive and representative data that better reflect global genetic diversity. Clinical trials conducted exclusively in European or broadly defined Asian populations may lack the external validity necessary to inform treatment across all patient groups. Moreover, categorizing Asian populations as a monolithic group fails to capture the genetic heterogeneity within this diverse demographic. Moving toward a more granular and comprehensive integration of pharmacogenomic data into clinical practice will be essential to optimizing drug therapy and realizing the full potential of precision medicine. Our findings reinforce the importance of more granular population stratification in the design and interpretation of pharmacogenetic research.

## 8. Future Directions

The observed variability in *CYP2C9* and *CYP2C19* allele frequencies across ethnic groups underscores the potential for differential drug responses among diverse populations. Such genetic differences can significantly influence therapeutic outcomes, especially for medications with narrow therapeutic indices, where even minor variations in drug metabolism may lead to toxicity or therapeutic failure. Current prescribing guidelines may be limited by evidence derived from clinical trials with insufficient representation of ethnic minority populations, potentially compromising their generalizability. Future clinical trials should prioritize inclusive recruitment strategies that span multiple geographic regions and demographic groups to enhance external validity and ensure equitable evidence generation. Moreover, the development of pharmacogenetic guidelines should explicitly integrate data from underrepresented population subgroups, enabling clinicians to make more informed assessments and population-specific prescribing decisions. By incorporating genetic diversity into both research and clinical practice, we can move closer to the goal of equitable and precision-based healthcare for all patients.

## Figures and Tables

**Figure 1 jpm-15-00274-f001:**
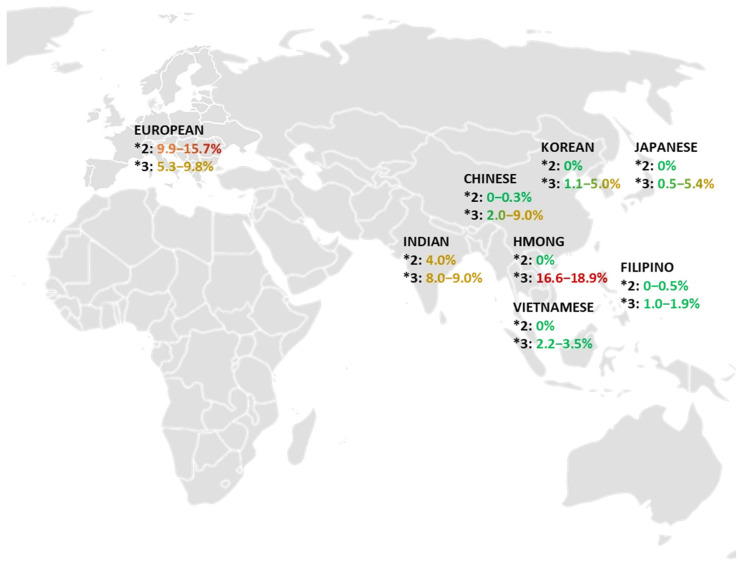
*CYP2C9*2* and **3* allele frequency in European and Asian population subgroups.

**Figure 2 jpm-15-00274-f002:**
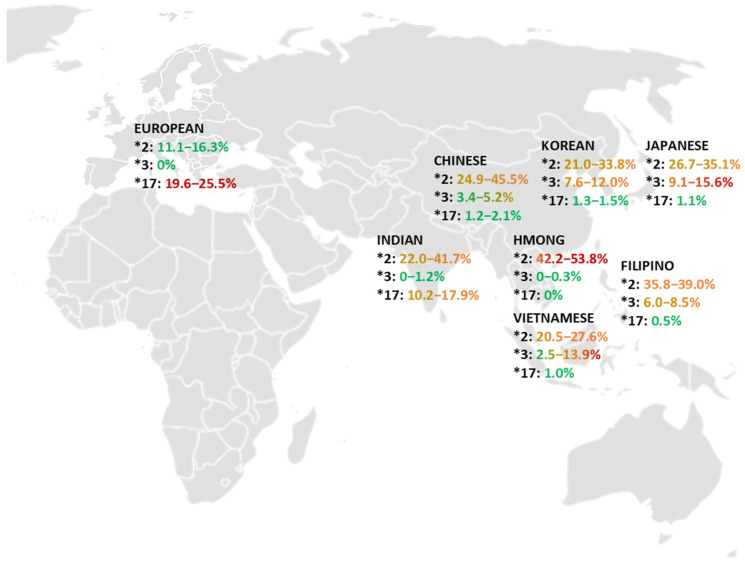
*CYP2C19**2, ***3, and ***17 allele frequency in European and Asian population subgroups.

**Table 1 jpm-15-00274-t001:** Drug label and clinical annotations for select CYP2C9 substrates.

Medication Name	*CYP2C9* PGx Label LOR	Approved Drug Label				Clinical PGx Guidelines	
		EMA	FDA	HC	SM	CPIC	DPWG
Siponimod	Testing Required	✓	✓	✓			✓
	Actionable PGx						
Celecoxib	Actionable PGx		✓	✓	✓	✓	
Dronabinol	Actionable PGx		✓				
Fosphenytoin	Actionable PGx		✓			✓	
	Informative PGx			✓			
Glimepiride	Actionable PGx			✓			
Glyburide	Actionable PGx			✓			
Losartan	Actionable PGx				✓		
Phenytoin	Actionable PGx		✓		✓	✓	✓
Warfarin	Actionable PGx		✓	✓		✓	✓
Meloxicam	Informative PGx		✓			✓	
Prasugrel	Informative PGx				✓		

LOR: Level of Recommendations; FDA: Food and Drug Administration; EMA: European Medicines Agency; HC: Health Canada; SM: Swissmedic; CPIC: Clinical Pharmacogenetics Implementation Consortium; DPWG: Dutch Pharmacogenomics Working Group.

**Table 2 jpm-15-00274-t002:** Drug label and clinical annotations for select CYP2C19 substrates.

Medication Name	*CYP2C19* PGx Label LOR	Approved Drug Labels				Clinical PGx Guidelines	
		EMA	FDA	HC	SM	CPIC	DPWG
Mavacamten	Testing Required	✓					
	Informative PGx		✓	✓			
Amitriptyline	Actionable PGx			✓		✓	✓
Carisoprodol	Actionable PGx		✓				
Citalopram	Actionable PGx		✓	✓	✓	✓	✓
Clobazam	Actionable PGx		✓	✓			
Clopidogrel	Actionable PGx		✓	✓	✓	✓	✓
	Informative PGx	✓					
Citalopram	Actionable PGx		✓	✓	✓	✓	✓
Escitalopram	Actionable PGx			✓	✓	✓	✓
	Informative PGx		✓				
Lansoprazole	Actionable PGx				✓	✓	✓
Pantoprazole	Actionable PGx		✓			✓	✓
	Informative PGx				✓		
Voriconazole	Actionable PGx				✓	✓	✓
	Informative PGx	✓	✓	✓			
Diazepam	Informative PGx		✓	✓			
Omeprazole	Actionable PGx				✓	✓	✓
	Informative PGx		✓				
Phenytoin	Informative PGx		✓				
Prasugrel	Informative PGx				✓		
Ticagrelor	Informative PGx	✓					
Dexlansoprazole	Actionable PGx				✓	✓	
	Informative PGx		✓	✓			
Esomeprazole	Actionable PGx				✓		
	Informative PGx		✓				

LOR: Level of Recommendations; FDA: Food and Drug Administration; EMA: European Medicines Agency; HC: Health Canada; SM: Swissmedic; CPIC: Clinical Pharmacogenetics Implementation Consortium; DPWG: Dutch Pharmacogenomics Working Group.

**Table 3 jpm-15-00274-t003:** *CYP2C9* diplotype–phenotype translation.

*CYP2C9* Diplotype	Activity Score	Phenotype
**1/*1*	2.0	Normal Metabolizer
**1/*2*	1.5	Intermediate Metabolizer
**1/*3*	1.0	Intermediate Metabolizer
**1/*5*	1.5	Intermediate Metabolizer
**1/*8*	1.5	Intermediate Metabolizer
**1/*11*	1.5	Intermediate Metabolizer
**2/*2*	1.0	Intermediate Metabolizer
**2/*3*	0.5	Poor Metabolizer
**2/*5*	1.0	Intermediate Metabolizer
**2/*8*	1.0	Intermediate Metabolizer
**2/*11*	1.0	Intermediate Metabolizer
**3/*3*	0.0	Poor Metabolizer
**3*5*	0.5	Poor Metabolizer
**3/*8*	0.5	Poor Metabolizer
**3/*11*	0.5	Poor Metabolizer
**5/*5*	1.0	Intermediate Metabolizer
**5/*8*	1.0	Intermediate Metabolizer
**5/*11*	1.0	Intermediate Metabolizer
**8/*8*	1.0	Intermediate Metabolizer
**8/*11*	1.0	Intermediate Metabolizer
**11/*11*	1.0	Intermediate Metabolizer

Source: PharmGKB (https://www.pharmgkb.org/page/cyp2c9RefMaterials (accessed on 19 June 2025)).

**Table 4 jpm-15-00274-t004:** *CYP2C19* diplotype–phenotype translation.

*CYP2C19* Diplotype	Activity Score	Phenotype
**1/*1*	n/a	Normal Metabolizer
**1/*2*	n/a	Intermediate Metabolizer
**1/*3*	n/a	Intermediate Metabolizer
**1/*17*	n/a	Rapid Metabolizer
**2/*2*	n/a	Poor Metabolizer
**2/*3*	n/a	Poor Metabolizer
**2/*17*	n/a	Intermediate Metabolizer
**3/*3*	n/a	Poor Metabolizer
**3*17*	n/a	Intermediate Metabolizer
**17/*17*	n/a	Ultrarapid Metabolizer

n/a: not applicable; Source: PharmGKB (https://www.pharmgkb.org/page/cyp2c19RefMaterials (accessed on 19 June 2025)).

**Table 5 jpm-15-00274-t005:** *CYP2C9* alleles frequencies in major populations and distinct subgroups.

Population	*CYP2C9* Allele Frequency (%)					References
	**2*	**3*	**5*	**8*	**11*	
Overall European (Range)	9.9–15.7	5.3–9.8				
○ Croatian	14.7	7.6				[27]
○ Danish	12.1	5.3				[28]
○ Norwegian	9.9	6.5				[28]
○ Romanian	11.3	9.3				[29]
○ Serbian	11.7	8.1				[30]
○ Spanish	15.6	9.8				[31]
○ Spanish, Northern Italian	15.7	7.8				[32]
Overall Asian (Range)	0–4.0	0.5–18.9	0	1.8	0–0.05	
○ Chinese	0–0.3	2.0–9.0	0	1.8	0.05	[33]
○ Filipino	0–0.5	1.0–1.9				[16,34]
○ Hmong	0	16.6–18.9				[35]
○ Indian	4.0	8.0–9.0				[36,37]
○ Japanese	0	0.5–5.4	0			[16,33,36,38,39]
○ Korean	0	1.1–5.0	0		0	[16,33,36,38]
○ Vietnamese	0	2.2–3.5				[33,40]

**Table 6 jpm-15-00274-t006:** *CYP2C19* alleles frequencies in major populations and distinct subgroups.

Population	*CYP2C19* Allele Frequency (%)			References
	**2*	**3*	**17*	
Overall European (Range)	11.1–16.3	0	19.6–25.5	
○ Croatian		0	23.9	[27]
○ Danish	15.0		20.1	[28]
○ German	15.2	0	25.5	[41]
○ Greek	13.1	0	19.6	[42]
○ Italian	11.1	0		[43]
○ Norwegian	15.2		22.0	[28]
○ Serbian	16.3		22.2	[30]
Overall Asian (Range)	20.5–53.8	0–15.6	0–17.9	
○ Chinese	24.9–45.5	3.4–5.2	1.2–2.1	[33,44,45,46,47]
○ Filipino	35.8–39.0	6.0–8.5	0.5	[48,49]
○ Hmong	42.2–53.8	0–0.3	0	[50]
○ Indian	22.0–41.7	0–1.2	10.2–17.9	[37,51,52,53,54]
○ Japanese	26.7–35.1	9.1–15.6	1.1	[33,47,55,56]
○ Korean	21.0–33.8	7.6–12.0	1.3–1.5	[33,48,57,58,59,60]
○ Vietnamese	20.5–27.6	2.5–13.9	1.0	[33,40,47,59]

## Data Availability

Not applicable.

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
