# Peer review of "Major Allele Frequencies in CYP2C9 and CYP2C19 in Asian and European Populations: A Case Study to Disaggregate Data Among Large Racial Categories"

_jpm, 2025, doi:10.3390/jpm15070274_

Round 1
Reviewer 1 Report
Comments and Suggestions for Authors
The authors are to be commended for addressing a highly relevant and timely topic in precision medicine and pharmacogenomics. The article effectively highlights the limitations of using broad racial categories, such as "Asian," and convincingly argues for the need to disaggregate data to prevent health disparities and optimize drug therapy. This is a strong and timely narrative review that makes an important contribution to the literature on pharmacogenomics and health disparities. Its core message is clear and well-supported.
Major Points
Study Type:
It is recommended that the authors specify the type of review conducted (e.g., "This narrative review aims to...") and adjust the text accordingly. This will help manage reader expectations.
Methods:
The authors could strengthen this section by adding:
More explicit inclusion and exclusion criteria (e.g., types of studies included/excluded, minimum sample size, etc.).
A brief description of how frequencies were calculated when not explicitly stated in the source articles (e.g., "calculated from available genotype counts").
Consideration of a small flow diagram (PRISMA-style, even if simplified) showing the number of articles found, screened, and included in the final analysis. This would add significant visual and methodological value.
Results:
Tables 3 and 4 could be improved by including the sample size (N) from each study or an overall N for each subgroup, not just the percentage. This would allow the reader to better gauge the number of individuals assessed to arrive at the presented frequency estimates.
Discussion:
The authors could briefly discuss why aggregating European subgroups (e.g., Croatians, Danes) might be less problematic (as suggested by the lower variability in their data) compared to aggregating Asian subgroups. This could be related to different migratory histories and greater genetic divergence among the included Asian populations.
The mention of variability within the Chinese population (line 226) is excellent. It would be even more impactful if the authors could cite a concrete example of how frequencies differ between, for instance, the Han and another minority group, using the data from references 37 and 39.
Minor Points
Line 42: The phrase "can alter the amino acid of the encoded protein" could be more precise as "can alter the amino acid sequence of the encoded protein."
Line 55: The abbreviation "HCSC" for Health Canada is uncommon. It is suggested to either spell out "Health Canada" or use the more common abbreviation "HC" for clarity.
Line 68: The phrase "aggregated into one single population" is redundant. "Aggregated into a single population" is sufficient.
Line 143: The statement "There was no data regarding European allele frequency for CYP2C9*5, CYP2C9*8, and CYP2C9*11 in the reviewed articles" is an important observation. This point could be moved to the Limitations section or emphasized here as an example of the lack of comprehensive data, even for European populations.
Line 238: There appears to be a minor confusion in the citation. The text states that a study on the Hmong population in the U.S. was used for CYP2C9 data and cites reference 53. However, reference 53 (Zhou et al., 2009) is a broad review on CYP polymorphisms, not a primary study on the Hmong in the U.S. The correct references for the Hmong data appear to be 27 and 28. The authors should verify this citation to ensure accuracy.
Author Response
The authors are to be commended for addressing a highly relevant and timely topic in precision medicine and pharmacogenomics. The article effectively highlights the limitations of using broad racial categories, such as "Asian," and convincingly argues for the need to disaggregate data to prevent health disparities and optimize drug therapy. This is a strong and timely narrative review that makes an important contribution to the literature on pharmacogenomics and health disparities. Its core message is clear and well-supported.
Major Points
Study Type:
It is recommended that the authors specify the type of review conducted (e.g., "This narrative review aims to...") and adjust the text accordingly. This will help manage reader expectations.
- Thank you for the comment. We emphasized the type of our review, “narrative review,” in lines 18 and 122
Methods:
The authors could strengthen this section by adding:
More explicit inclusion and exclusion criteria (e.g., types of studies included/excluded, minimum sample size, etc.).
- Thank you for your suggestions. Though our manuscript is a narrative review, we didn’t apply strict inclusion or exclusion criteria. However, we added a more detailed description of inclusion and exclusion criteria to the method section. Please see line 131-146
A brief description of how frequencies were calculated when not explicitly stated in the source articles (e.g., "calculated from available genotype counts").
- Thank you for the suggestion. We added additional description in line 137-139
Consideration of a small flow diagram (PRISMA-style, even if simplified) showing the number of articles found, screened, and included in the final analysis. This would add significant visual and methodological value.
- This is a narrative review that did not warrant the PRISMA flow diagram of a systematic review. We appreciate the suggestion and will be mindful of tracking the number of articles in future systematic reviews.
Results:
Tables 3 and 4 could be improved by including the sample size (N) from each study or an overall N for each subgroup, not just the percentage. This would allow the reader to better gauge the number of individuals assessed to arrive at the presented frequency estimates.
- Thank you for the suggestion. The current tables in the manuscript were streamlined to make the data more user-friendly. We appreciate your suggestion and have added our itemized table with sample sizes per reviewed article as supplementary files as Table S7 and Table S8.
Discussion:
The authors could briefly discuss why aggregating European subgroups (e.g., Croatians, Danes) might be less problematic (as suggested by the lower variability in their data) compared to aggregating Asian subgroups. This could be related to different migratory histories and greater genetic divergence among the included Asian populations.
- Thank you for the comment. We added a brief explanation in lines 244-252.
The mention of variability within the Chinese population (line 226) is excellent. It would be even more impactful if the authors could cite a concrete example of how frequencies differ between, for instance, the Han and another minority group, using the data from references 37 and 39.
- Thank you for the comment. We added explanation regarding a distinct subgroup difference from reference 61 in lines 244-252.
Minor Points
Line 42: The phrase "can alter the amino acid of the encoded protein" could be more precise as "can alter the amino acid sequence of the encoded protein."
Line 55: The abbreviation "HCSC" for Health Canada is uncommon. It is suggested to either spell out "Health Canada" or use the more common abbreviation "HC" for clarity.
Line 68: The phrase "aggregated into one single population" is redundant. "Aggregated into a single population" is sufficient.
Line 143: The statement "There was no data regarding European allele frequency for CYP2C9*5, CYP2C9*8, and CYP2C9*11 in the reviewed articles" is an important observation. This point could be moved to the Limitations section or emphasized here as an example of the lack of comprehensive data, even for European populations.
Line 238: There appears to be a minor confusion in the citation. The text states that a study on the Hmong population in the U.S. was used for CYP2C9 data and cites reference 53. However, reference 53 (Zhou et al., 2009) is a broad review on CYP polymorphisms, not a primary study on the Hmong in the U.S. The correct references for the Hmong data appear to be 27 and 28. The authors should verify this citation to ensure accuracy.
- All minor points were updated per suggestions. Thank you!
Reviewer 2 Report
Comments and Suggestions for Authors
The authors present the work titled "Major Allele Frequencies in CYP2C9 and CYP2C19 in Asian 2 and European Populations: A Case Study to Disaggregate Data 3 Among Large Racial Categories" with the aim of exploring the differences in CYP2C9 and CYP2C19 allele frequencies among European and Asian population subgroups.
The study is clear, although not complex; the objective is well-defined, as is the selection of populations based on the relevance of the genes under study.
The English writing is generally clear, with no major issues except for some minor stylistic corrections, for example Limitations section (line 329).
Some improvement opportunities include:
1. Please indicate that “CYP2C” stands for Cytochrome P450 family 2 subfamily C, and so.
2. Some sections of the text require additional references, particularly in the introduction (page 3, first paragraphs).
3. It would be useful to present the results in Tables 3 and 4 including data from otehr regions (Africa, the Americas, other parts of Asia or Europe), allowing for comparison across regions, even within Europe and Asia in other works. Such comparisons should be based on published data or databases related to gene frequency. An example of a suitable reference:
https://www.researchgate.net/figure/Distributions-of-CYP2C19-CYP2C9-and-CYP2D6-star-allele-frequencies-among-global_fig3_356535597
4. Consider adding a summary table or diagram showing how the different alleles of the studied genes correlate with metabolizer phenotypes (poor, intermediate, rapid, ultra-rapid), and include allele frequencies by region. Although this is mentioned in the text, a summarized graphic would enhance the emphasis on these aspects.
5. The allele frequency maps in the figures could be improved by using a heatmap, where color intensity represents frequency. This would make comparisons more visually clear. If a map is not used, the corresponding table should clearly reflect this information. The figures do not need to be removed, but rather enhanced.
6. Although not the main focus of this study, it would be relevant to expand the discussion or introduce the importance of genotyping through various molecular strategies. It is also important to highlight the need for comprehensive representation of countries and geographic regions in databases. This would help emphasize the ongoing need for molecular analyses to identify predominant alleles in different populations, especially in a highly globalized and mobile world.
Author Response
The authors present the work titled "Major Allele Frequencies in CYP2C9 and CYP2C19 in Asian 2 and European Populations: A Case Study to Disaggregate Data 3 Among Large Racial Categories" with the aim of exploring the differences in CYP2C9 and CYP2C19 allele frequencies among European and Asian population subgroups.
The study is clear, although not complex; the objective is well-defined, as is the selection of populations based on the relevance of the genes under study.
The English writing is generally clear, with no major issues except for some minor stylistic corrections, for example Limitations section (line 329).
- Thank you for thorough review. We updated font and size to correct stylistic errors. Corrected other formatting errors throughout the paper in bold font, clarifying abbreviations, and referencing Figures.
Some improvement opportunities include:
- Please indicate that “CYP2C” stands for Cytochrome P450 family 2 subfamily C, and so.
- Added family-subfamily-member information in the Background section when first introducing the CYP2C9 and CYP2C19 enzymes in lines 37-39.
- Some sections of the text require additional references, particularly in the introduction (page 3, first paragraphs).
- Thank you. We added more contemporary references to substantiate the mentioned paragraphs in question. Added references to page 3 first paragraph.
- It would be useful to present the results in Tables 3 and 4 including data from other regions (Africa, the Americas, other parts of Asia or Europe), allowing for comparison across regions, even within Europe and Asia in other works. Such comparisons should be based on published data or databases related to gene frequency. An example of a suitable reference:
https://www.researchgate.net/figure/Distributions-of-CYP2C19-CYP2C9-and-CYP2D6-star-allele-frequencies-among-global_fig3_356535597
- Thank you for the feedback but the comparison to other continents would not fit the scope of the paper, which was a comparison of European and Asian populations. We will consider the data from other regions in the shared reference for future reviews with an expanded scope.
- Consider adding a summary table or diagram showing how the different alleles of the studied genes correlate with metabolizer phenotypes (poor, intermediate, rapid, ultra-rapid), and include allele frequencies by region. Although this is mentioned in the text, a summarized graphic would enhance the emphasis on these aspects.
- Thank you for the suggestions. Summary tables for the CYP2C9 and CYP2C19 diplotype- phenotype translation were added in both Table 3 and Table 4.
- The allele frequency maps in the figures could be improved by using a heatmap, where color intensity represents frequency. This would make comparisons more visually clear. If a map is not used, the corresponding table should clearly reflect this information. The figures do not need to be removed, but rather enhanced.
- This was a great suggestion to utilize visual learning. For Figures 1 and 2, the font colors were updated to resemble a heatmap using a color gradient to emphasize the difference in the allele frequency percentages. We are unable to generate a heatmap based on our available resources. In addition, the Hmong population would be difficult to assign to a specific region/country as a refugee/displaced population. Figure 1 corresponds to Table 5 and Figure 2 corresponds to Table 6.
- Although not the main focus of this study, it would be relevant to expand the discussion or introduce the importance of genotyping through various molecular strategies. It is also important to highlight the need for comprehensive representation of countries and geographic regions in databases. This would help emphasize the ongoing need for molecular analyses to identify predominant alleles in different populations, especially in a highly globalized and mobile world.
- Thank you for the comment. We agree this is a good addition to include in our paper. Please see lines 368-374.
Reviewer 3 Report
Comments and Suggestions for Authors
The submitted manuscript presents a valuable review of a clinically significant topic — the impact of genetically determined variability in CYP2C9 and CYP2C19 enzyme activity on the pharmacokinetics of various drugs. The authors have carefully selected the literature, including both population-level data and specific therapeutic implications. The text is clearly written, well-structured, and accessible to readers with medical or pharmaceutical backgrounds, both specialists in clinical and basic sciences.
A particular strength of the review is the well-organized summary of genotype–phenotype interactions for drugs commonly used in clinical practice. The manuscript is of high educational value and may serve as a useful reference for clinicians and pharmacists interested in pharmacogenetics.
It is worth noting that some medicines, important becouse of wide usage in clinical medicine such as carbamazepine is not discussed in detail — due to the predominant role of CYP3A4 in its metabolism — it remains a classic example of a drug where therapeutic drug monitoring (TDM) and pharmacogenetic considerations have provided tangible clinical benefits. We already have publications summarizing more than 20 years of experience with monitored carbamazepine therapy (as well as with TDM classics such as digoxin), making it an important reference point for the entire concept of individualized pharmacotherapy. In my opinion TDM in that cases in selected populations seem to be basic of personalised therapy.
The value of the manuscript could be further enhanced by including a graphical representation of the population-based distribution of key CYP2C9 and CYP2C19 alleles. A geographic map, with countries color-coded according to the frequency of relevant polymorphisms, would help illustrate the global relevance of this topic and support clinical decision-making in multiethnic populations.
This is a well-prepared and up-to-date manuscript, based on reliable literature and well-documented facts. After minor editorial enhancements — particularly regarding geographic allele distribution and adding backgruoun for TDM personalised therapy
Author Response
The submitted manuscript presents a valuable review of a clinically significant topic — the impact of genetically determined variability in CYP2C9 and CYP2C19 enzyme activity on the pharmacokinetics of various drugs. The authors have carefully selected the literature, including both population-level data and specific therapeutic implications. The text is clearly written, well-structured, and accessible to readers with medical or pharmaceutical backgrounds, both specialists in clinical and basic sciences.
A particular strength of the review is the well-organized summary of genotype–phenotype interactions for drugs commonly used in clinical practice. The manuscript is of high educational value and may serve as a useful reference for clinicians and pharmacists interested in pharmacogenetics.
It is worth noting that some medicines, important because of wide usage in clinical medicine such as carbamazepine is not discussed in detail — due to the predominant role of CYP3A4 in its metabolism — it remains a classic example of a drug where therapeutic drug monitoring (TDM) and pharmacogenetic considerations have provided tangible clinical benefits. We already have publications summarizing more than 20 years of experience with monitored carbamazepine therapy (as well as with TDM classics such as digoxin), making it an important reference point for the entire concept of individualized pharmacotherapy. In my opinion TDM in that cases in selected populations seem to be basic of personalized therapy.
- Thank you for the suggestions and comments. We added two new paragraphs in the Discussion section that discussed the importance TDM in relation to PGx. Please see lines 333-353
The value of the manuscript could be further enhanced by including a graphical representation of the population-based distribution of key CYP2C9 and CYP2C19 alleles. A geographic map, with countries color-coded according to the frequency of relevant polymorphisms, would help illustrate the global relevance of this topic and support clinical decision-making in multiethnic populations.
- This was a great suggestion to utilize visual learning. For Figures 1 and 2, the font colors were updated to resemble a heatmap using a color gradient to emphasize the difference in the allele frequency percentages. We are unable to generate a heatmap based on our available resources. In addition, the Hmong population would be difficult to assign to a specific region/country as a refugee/displaced population. Alternatively, we significantly expanded both tables 1 and 2 to reinforce how different clinical guidelines and regulatory agencies issue their pharmacogenomics recommendations for certain drugs which can be impacted by the population allele frequencies.
This is a well-prepared and up-to-date manuscript, based on reliable literature and well-documented facts. After minor editorial enhancements — particularly regarding geographic allele distribution and adding background for TDM personalized therapy.
- Thank you for your time to review our manuscript.
Round 2
Reviewer 2 Report
Comments and Suggestions for Authors
authors addressed all points 😉